# Design of a Movable Tensegrity Arm with Springs Modeling an Upper and Lower Arm

**Kihiro Kawahara** , **Duk Shin** and **Yuta Ogai** *

Department of Electronics and Information Technology, Graduate School of Engineering,
Tokyo Polytechnic University, 1583 Iiyama, Atsugi 243-0297, Kanagawa, Japan
* Correspondence: ogai@t-kougei.ac.jp

**Abstract:** Tensegrity is a structure consisting of rigid bodies and internal tensile members, with no contact between the rigid bodies. The model of an arm with a tensegrity structure is not movable as it is, but we believe that it can be made movable and flexible by incorporating springs. We developed an arm that incorporates springs in the arm's tensile members by extending the model of an arm with a tensegrity structure. Then, as an evaluation of the developed arm, we measured the ranges of motions and the forces required for that motion. We also developed a mechanism that allows the arm to bend and extend. We believe that this method of making the tensegrity arm controllable by incorporating springs will be useful in the development of flexible robotic arms for caregiving using robots and other applications.

**Keywords:** tensegrity; robot arm; soft robotics





## 1. Introduction

The importance of the body in intelligence has long been the focus of attention [1,2], but recent developments in deep learning technology and other factors have further accelerated the development of systems that incorporate AI in robots. However, just as there are differences in the way humans use their bodies, robots can be considered to have different degrees of embodiment. Even if an actual robot is used, if the relationship between the robot and the world is simple, the degree of the embodiment can be considered low. Conversely, if the relationship between the agent and the world is complex, even in a simulation, the degree of the embodiment is high. We do not yet have an idea for quantifying the degree of embodiment, but we believe that the unpredictability of changes in behavior when the environment changes is a candidate. Ogai et al. claimed that they were able to evolve the behavior of a neural-network-equipped agent, moving in two-dimensional space, to acquire chaotic behavior with respect to selection by goal location [3]. Research is also being conducted on the use of actual equipment to construct interactions between the body and the environment [4]. Industrial robot arms are usually modeled in a computationally tractable structure using methods such as homogeneous transformation matrices. However, in this paper, we consider structures that are able to use the body as a rich computational resource, and we believe that tensegrity is a useful method for obtaining such structures.

Tensegrity is a structure consisting of rigid bodies and internal tensile members, with no contact between the rigid bodies. An example of a Tensegrity structure is shown in Figure 1. The name "tensegrity" has been given by R. Buckminster Fuller [5,6]. R. E. Skelton and M. C. de Oliveira defined a tensegrity configuration of rigid bodies in detail, as follows: [5].

In the absence of external forces, let a set of rigid bodies in a specific configuration have torqueless connections (e.g., via frictionless ball-joints). Then this configuration forms a tensegrity configuration if the given configuration can be stabilized by some set of internal tensile members, i.e., connected between the rigid bodies. The configuration is

not a tensegrity configuration if no tensile members are required and/or no set of tensile members exist to stabilize the configuration.

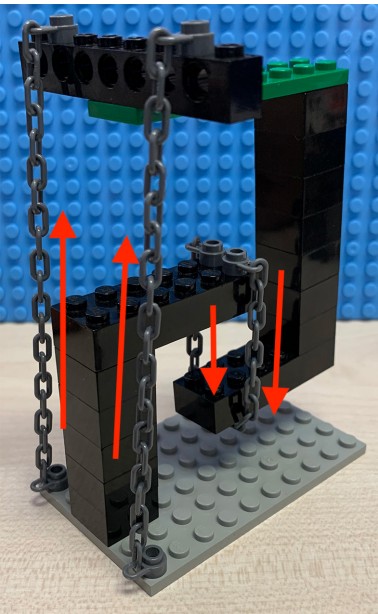

**Figure 1.** An example of a tensegrity structure formed with LEGO. The blocks are rigid bodies, and the chains are tensile members. The red arrows indicate the directions of the forces on the chains.

The space inside the tensegrity structure can be used widely and made larger with lighter materials. First, tensegrity was used as the structure of artworks [7]. It has since spread to the architectural field. For example, it is also used for the Kurilpa Bridge in Australia [8] (Figure 2). Various studies have also been conducted on the classification and analysis of tensegrity structures [5,9–11]. R. E. Skelton and M. C. de Oliveira defined a Class 1 tensegrity system as a tensegrity configuration that has no points of contact between its rigid bodies [5]. In particular, they performed a detailed analysis of the type of Class 1 tensegrity system in which the nodes, that is, the joints between the tensile and compression members, are located only at the ends of the compression members.

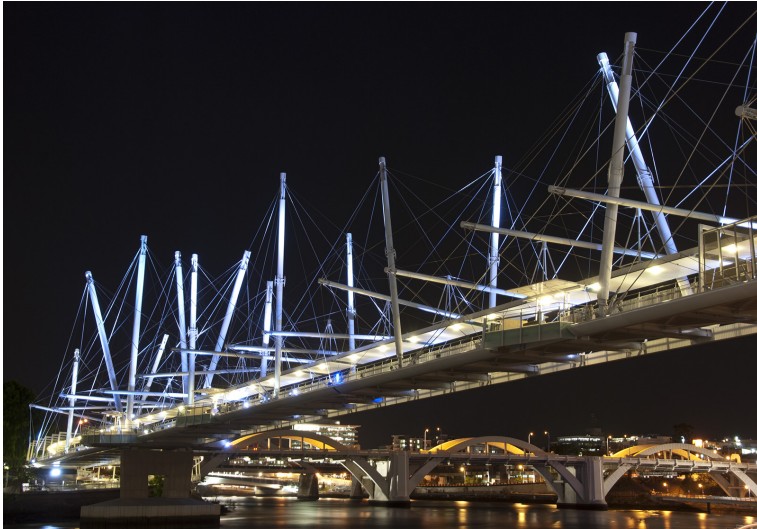

**Figure 2.** The Kurilpa Bridge in Brisbane, Australia, has a tensegrity structure [12].

It is also expected to have robustness and tolerance to forces applied from any direction and is being studied as a method of constructing robots [13–16]. There have been several studies on stacking tensegrity to create arms [17,18].

The similarities between tensegrity and human body structures have also been noted [19]. For example, D. E. Ingber wrote as follows: [20].

In other words, in the complex tensegrity structure inside every one of us, bones are the compression struts, and muscles, tendons and ligaments are the tension-bearing members.

S. M. Levin proposes the term biotensegrity for the structure of bone as a floating compressive material and fascia stretched between the compressive materials [21,22]. Some studies have constructed the structure of the arm by tensegrity [23,24].

F. Iida's research focused on how springs enable a robot to adapt to its environment [25]. Pitti et al. have coupled chaos to the system developed by Iida to allow the system to explore movement patterns autonomously [26]. The springs allow them to acquire complex movement patterns that are explored by chaos. Pitti et al. are also studying the structure of the leg, which is gently moved by the McKibben air muscle [27].

We consider that by inserting a spring in a tensile member, it would be possible to construct a structure that is flexible, stores force inside, and has dynamics. The flexibility provided by the springs in the arms is expected to allow the arm to be adaptable and to generate force instantaneously. For example, if compression materials are directly connected at joints, the joints are usually weak, and the compression materials may break at the joints when subjected to a strong impact. However, with a tensegrity structure, the tensile material would be adaptive by absorbing the shock, making it more resistant to impact. We also believe it is possible to quickly move the compression member by loosening only one side of the tensile member from both sides connected to the compression member, which are pulled at the same time. Such a structure would have characteristics similar to those of the human body. The skeleton of the compression members of tensegrity could also be made to resemble the human body, and through the process of developing such a skeleton, our understanding of how the human body moves could be improved. Such development would also lead to the development of humanoids with human-like movements. Air muscles can be controlled on their own, but springs are usually available with higher spring constants for the same size. There have been studies of tensegrity and spring models of the spinal cord [28], and we believe that springs can be used in tensegrity models of the arm as well. We also believe that this structure will lead to the study of arms that can compute as a physical reservoir computer [29].

In this case, however, the length of the tensile member is variable, making it difficult to determine the initial length of the tensile member. Therefore, this study explores how to design a tensegrity arm when springs are included in the tensile member. The range of motion and lifting ability of the developed arm when moved will also be evaluated. Biotensegrity considers fascia as a tensile material [21,22], but since it is difficult to realize the actual complex structure of fascia, we propose here an abstracted tensegrity model with a small number of parts. A detailed construction of the model, in which the upper and lower arms are modeled to resemble the structure of the human body closely, and mobility is provided by springs, has not been studied in previous research. We believe that research on how to construct a tensegrity arm that closely resembles the structure of the human body will lead to future research on human movements, such as sports, and human–robot interactions, such as caregiving using robots.

In Section 2, the structure of the developed tensegrity arm and how to determine the length of the tensile members, measure the range of motion, and actuate the arm with electric winches will be described. In Section 3, the length of tensile members of the tensegrity arm obtained through trial and error, the arm's range of motion, and the results of the operation of the arm are reported. In Section 4, based on the results, the relationship of the structure to the human body and future prospects are discussed. Finally, the paper is summarized in Section 5.

## 2. Materials and Methods

### 2.1. Developed Tensegrity Arm

We based our tensegrity model of an arm on the Tensegrity Research Group's website [24], and modified the model so that springs could be used as part of the tensile member. Our developed tensegrity model is shown as an approximate 3d CAD model in Figures 3 and 4a,c. Figure 4a shows the model from the side, and Figure 4b shows the human arm in the orientation corresponding to Figure 4a.

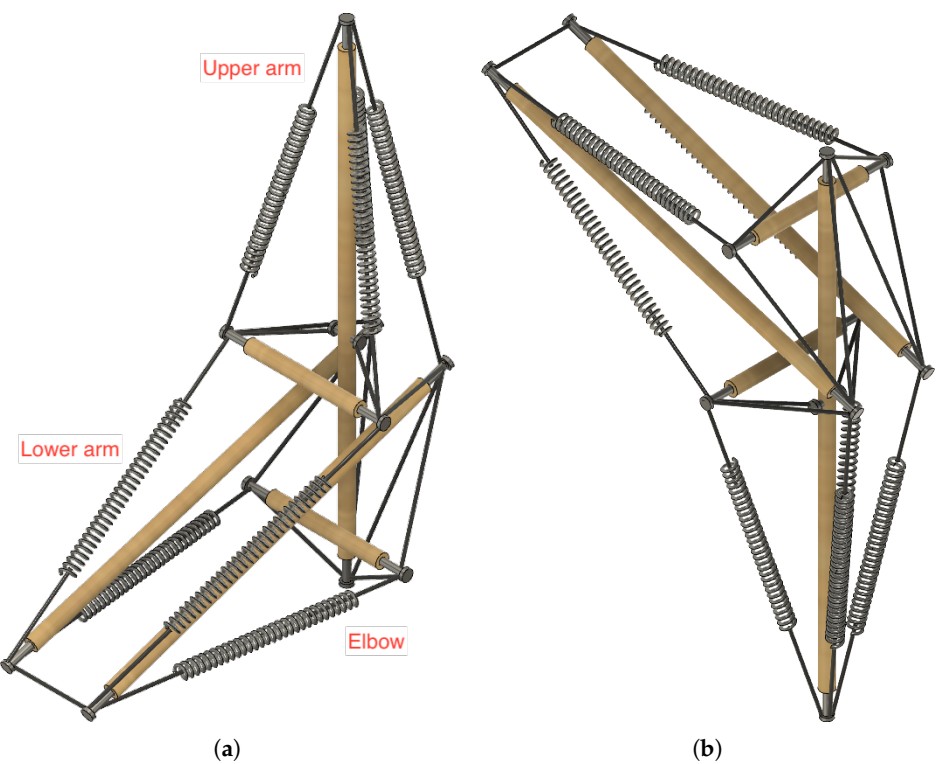

(**a**)    (**b**)

**Figure 3.** Approximate 3D CAD model of the developed tensegrity arm. Figure (**a**) shows the name of each location. Figure (**b**) shows the model in Figure (**a**) upside down.

The developed tensegrity arm is shown in Figure 5, and the materials for the arm are listed in Tables 1 and 2. In Figure 5, the area on the left is considered the upper arm, and the area on the right is considered the lower arm. The screws attached to the end of each wooden stick protrude by about 2 cm. As shown in Figures 6–8, each tensile member is numbered.

Since the compression members do not interfere with each other in this model, it is a Class 1 tensegrity structure [5]. However, as will be shown later in Figure 8, this is not a typical Class 1 tensegrity structure because there are connection nodes from the middle of the compression members to the tension members. These nodes were also in the original model.

**Table 1.** The sticks of the materials for the developed arm. "Position" is shown in Figure 4a.

| Position | Material | Size | Quantity |
|---|---|---|---|
| R1 | Round wooden stick | $\phi$ 1.2 cm, length 40 cm | 1 |
| R2 | Round wooden stick | $\phi$ 1.2 cm, length 30 cm | 2 |
| R3 | Round wooden stick | $\phi$ 1.2 cm, length 10 cm | 2 |

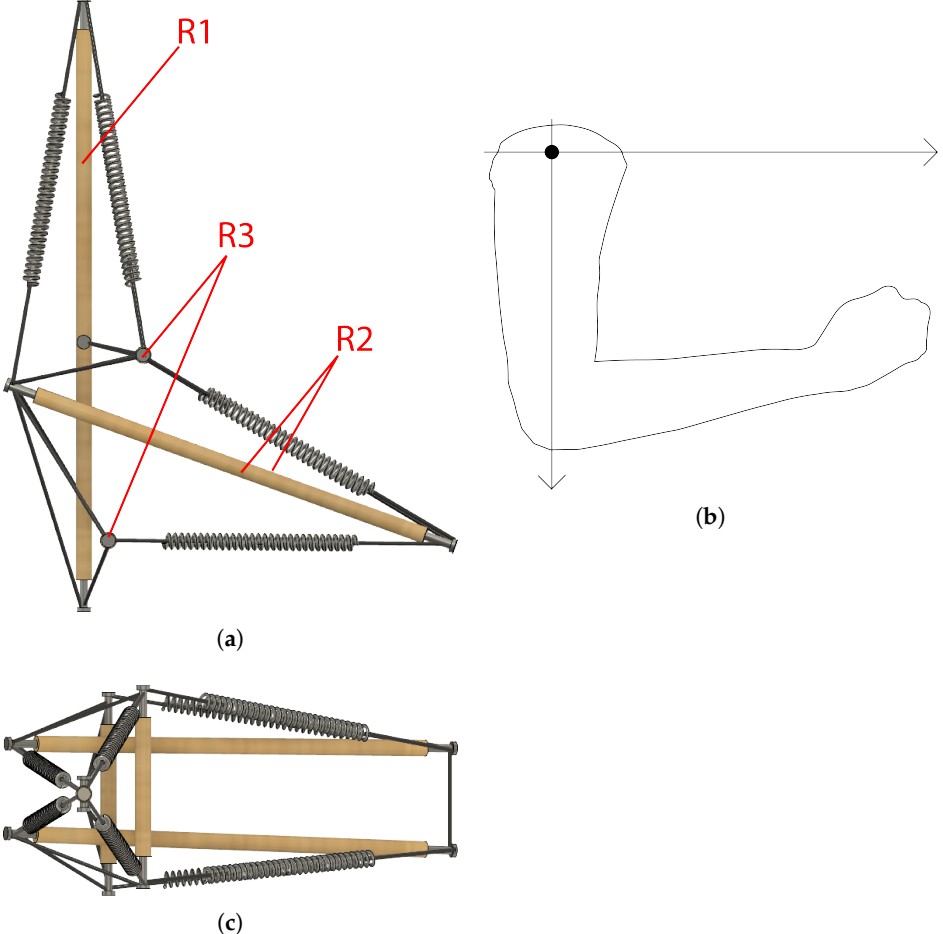

**Figure 4.** The correspondence between the developed tensegrity arm and a human arm. Figure (**a**) shows the model from the side and the name of each stick. Figure (**b**) shows the human arm in the orientation corresponding to Figure (**a**). Figure (**c**) shows the model from above.

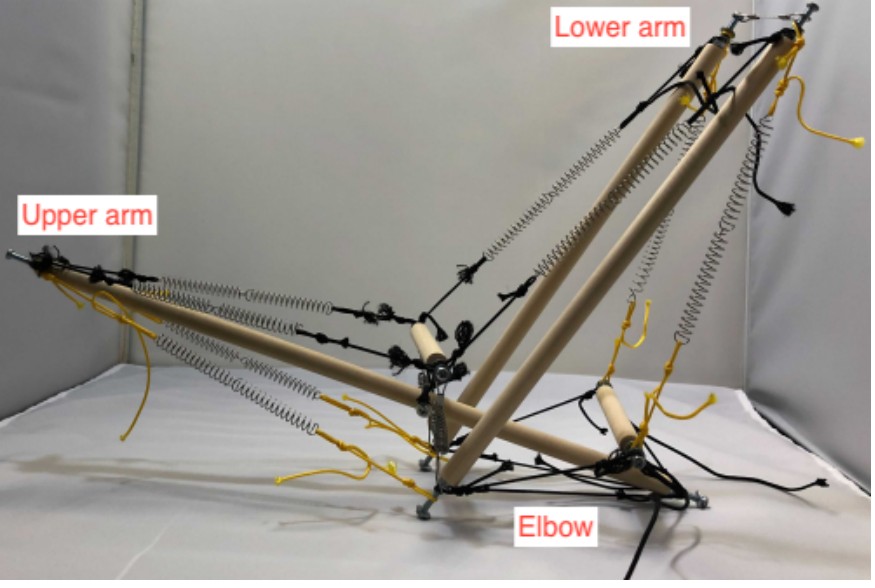

**Figure 5.** The developed tensegrity arm. The left side is the upper arm, the right side is the lower arm, and the center is the elbow.

**Table 2.** The materials other than the sticks for the developed arm.

| Material | Detail | Quantity |
| --- | --- | --- |
| Polyester string | $\phi$ 0.15 cm | 24 |
| Pan head screw | M4, pitch 0.07 cm, length 4 cm | 10 |
| Hexagonal nut | M4, pitch 0.07 cm, thickness 0.5 cm | 10 |
| Flat washer | M4, thickness 0.1 cm | 20 |
| Compression spring | natural length 5 cm, spring constant 0.985 | 16 |
| Wire | $\phi$ 0.08 cm | 5 |

Since it is difficult to know the forces applied to the tensile members previously, we obtained the lengths of the springs of the tensile materials by trial and error. The policy for trial and error is as follows:

1. The longer tensile materials (from No. 1 to No. 8 in Figures 6 and 7) were made about 30 cm long by two springs and strings.
2. The short tensile materials around the elbow (from No. 9 to No. 18 in Figure 8) were made about 15 cm long by a string.
3. After assembly, the slack in each string was contracted and adjusted so that the entire piece was taut.
4. The total balance was adjusted through trial and error.

The force on the spring is obtained by Equation (1), where $F$ is the force, $k$ is the spring constant, $x$ is the natural length of the spring, and $x'$ is the length of the spring when extended.

$$F = k(x' - x) \tag{1}$$

The screws holding the ends of the strings of No.17 and No.18 in Figure 8 have a screw hole drilled in the middle of the 40 cm compression material. The hole is located 17.2 cm from the elbow side of the compression material.

We modified the joints from the original model to make them stronger and the length of the strings easier to adjust (Figure 9). In the original model, the strings were cramped between gaps in the compression materials, but these are now cramped between screws, washers, and nuts.

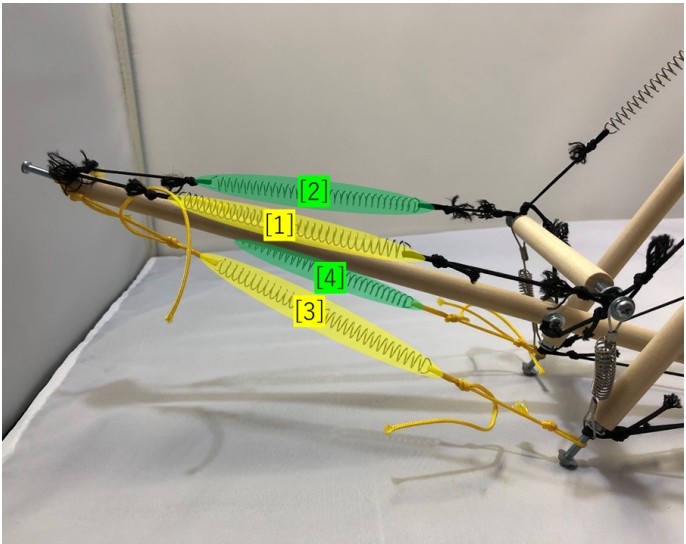

**Figure 6.** The numbers of the tensile members in the upper arm. In this figure, the tensile member inside the elbow at the front of the upper arm is No. 1; the member inside the elbow at the back is No. 2; the member outside the elbow at the front is No. 3; and the member outside the elbow at the back is No. 4. They are color-coded yellow and green to make them easier to see.

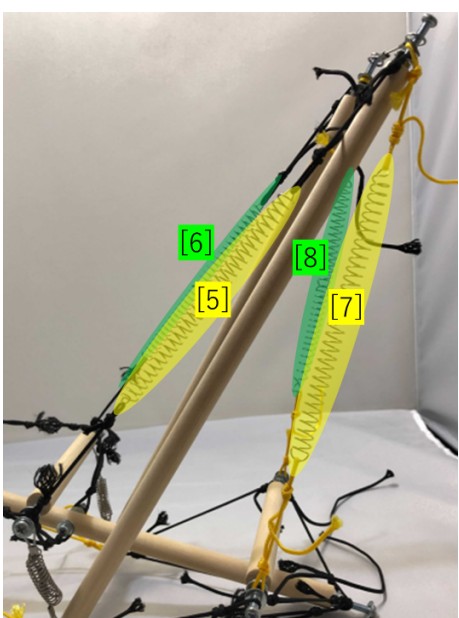

**Figure 7.** The numbers of the tensile members in the lower arm. The tip of the two sticks is connected by No. 19 wire shown in Figure 8. In this figure, the tensile member inside the elbow at the front of the lower arm is No. 5; the member inside the elbow at the back is No. 6; the member outside the elbow at the front is No. 7; and the member outside the elbow at the back is named No. 8. They are color-coded yellow and green to make them easier to see.

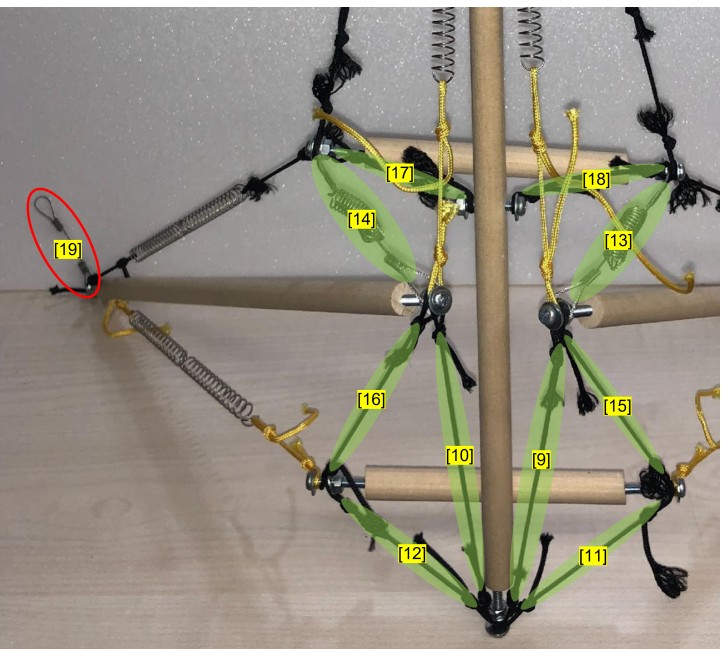

**Figure 8.** The numbers of tensile materials around the elbow. The tensile member leading from the elbow end of the upper arm to the elbow end of the right lower arm is No. 9; the tensile member leading to the elbow end of the left lower arm is No. 10; the tensile member leading to the right side of the outer elbow compression member is No. 11; the tensile member leading to the left side of the outer elbow compression member is No. 12; the tensile member leading from the right side of the inner compression member to the right lower arm is No. 13; and the tensile member leading from the left side of the inner compression member to the left lower arm is No. 14; the tensile member leading from the right side of the outer compression member to the right lower arm is No. 15; the tensile member leading from the left side of the outer compression member to the left lower arm is

No. 16; the tensile member leading from the screw in the center of the upper arm to the left side of the inner elbow compression member is No. 17; the tensile member leading to the right side is No. 18; and finally, the tensile member connecting the tips of the left and right lower arms is No. 19. In this figure, No. 19 at the end of the lower arm is disconnected, and the lower arm is open. No. 19 uses wire instead of string for convenience of removal. In No. 13 and No. 14, the ends of the springs are connected to the rigid body on the opposite side by wires. However, the springs hardly move. Therefore, functionally they are almost the same as strings.

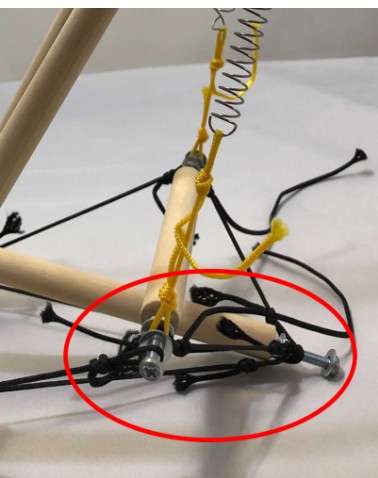

**Figure 9.** Joints between compression and tensile materials. The tensile member indicated by the red circle is No. 12.

### 2.2. Measurement of Range of Motion for the Tip of the Lower Arm

We measured the upward and downward ranges of motion for the lower arm of the developed arm (Figure 10). With the lower arms down, the upper arm was fixed vertically to the ceiling. A string was attached to the tip of the lower arm using pulleys on the ceiling, and the pulleys allowed the force to be applied by weight in each of the vertical directions (Figures 11 and 12). Weights were added every 500 g, and measurements were taken after the arm stopped. A ruler was placed vertically behind the arm and photographed from the front to measure the vertical position of the tip. The height of the tip with no weight was 16 cm from the base of the device.

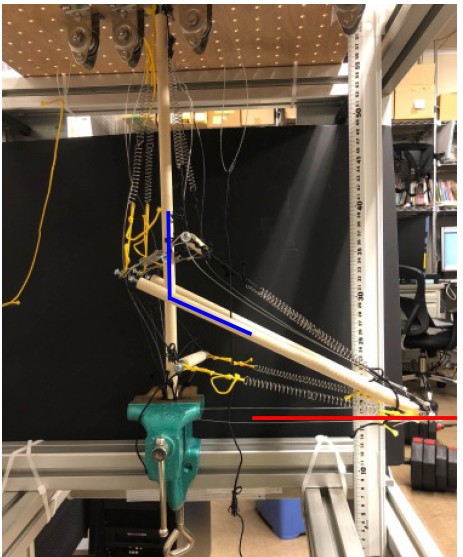

**Figure 10.** The environment to measure the range of motion of the tip. The red line is the height of the tip. The angle formed by the blue lines was measured.

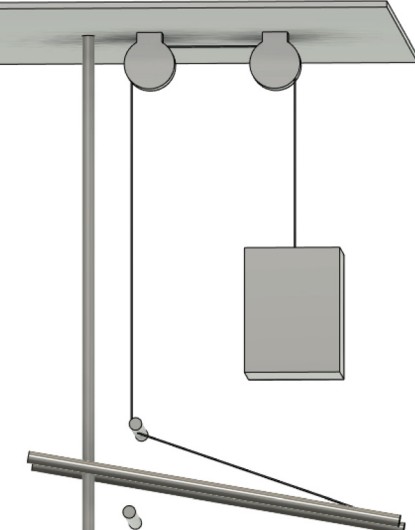

**Figure 11.** The path of the pulley string in the upward movement. From the tip of the lower arm, the string is passed through the short compression material inside the joint to apply the force of the weight. The weight is not in the position shown in this figure. The weight is in a position below the lower arm by extending the string so that it does not contact the arm.

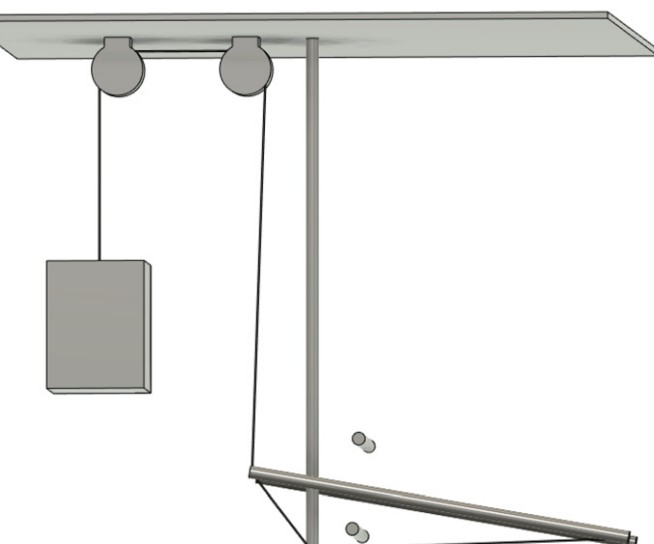

**Figure 12.** The path of the pulley string in the downward movement. The string is passed from the tip to the end of the compression material on the elbow side of the upper arm, and then to the end of the compression material on the elbow side of the lower arm.

### 2.3. Lifting Ability

We examined whether the arm could lift an object. For the case of a 500 g weight attached to the tip of the lower arm, we measured how the height of the tip changed as other weights were attached in 500 g increments to make the tip move in the upward direction.

### 2.4. Control by Electric Winches

We verified that the arm could be controlled using a device to pull a string. The strings were attached to the same position as shown in Section 2.2 and pulled with two electric winches to check their movement (Figure 13). One winch was an upward movement, and the other was a downward movement.

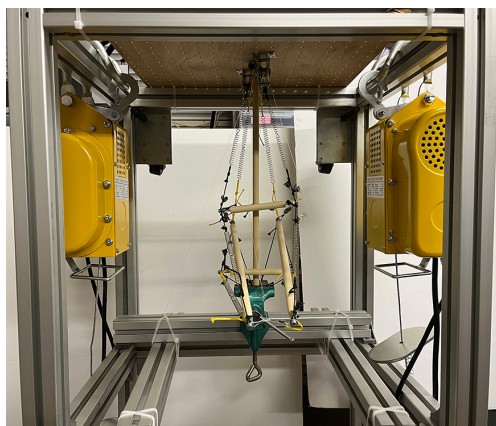

**Figure 13.** The environment to control the arm with the two electric winches. The tip of the arm can be pulled by an electric winch through the paths of the pulley strings shown in Figures 11 and 12.

## 3. Results

### 3.1. Lengths of Strings and Springs

By adjusting the lengths of the strings through trial and error, we were able to configure the tensegrity, arm as shown in Figure 5. The total lengths, the lengths of the two springs section, and the lengths of strings of the tensile materials from No. 1 to No. 8 after the trial and error adjustments are shown in Table 3. The lengths of the tensile materials from No. 9 to No. 19 are shown in Table 4. Since the average length of the springs was 14.23 cm, and the natural length of the two springs combined was 10 cm, the average force of 4.16 N was applied to each spring from Equation (1).

**Table 3.** The total length, the length of the two springs section, and the length of one string of the tensile member from No.1 to No.8. These tensile members consist of two springs with strings attached to each end.

| Tensile Member No. | Total Length (cm) | Two Springs Length (cm) | One String Length (cm) |
|---|---|---|---|
| 1 | 29.2 | 13.4 | 7.90 |
| 2 | 27.5 | 13.5 | 7.00 |
| 3 | 31.2 | 14.1 | 8.55 |
| 4 | 30.7 | 13.7 | 8.50 |
| 5 | 29.0 | 14.3 | 7.35 |
| 6 | 29.8 | 14.2 | 7.80 |
| 7 | 26.9 | 15.5 | 5.70 |
| 8 | 26.7 | 15.1 | 5.80 |

**Table 4.** The lengths of tensile members for strings for members No. 9 to No. 19.

| Tensile Member No. | Length (cm) |
|---|---|
| 9 | 15.4 |
| 10 | 15.5 |
| 11 | 8.8 |
| 12 | 8.8 |
| 13 | 8.9 |
| 14 | 8.8 |
| 15 | 11.0 |
| 16 | 11.6 |
| 17 | 6.5 |
| 18 | 7.2 |
| 19 | 6.9 |

### 3.2. Measurement of Range of Motion for the Tip of the Lower Arm

3.2.1. Upward Range of Motion

Figure 14 shows the tip pulled up with a weight of 5000 g. Based on the angle of the string pulled, we consider that the angle of the lower arm in Figure 14 will be the limit. Figure 15 shows the graph of the height of the tip, and Figure 16 shows the graph of the angle between the upper arm and the lower arm when it was pulled up by the weights.

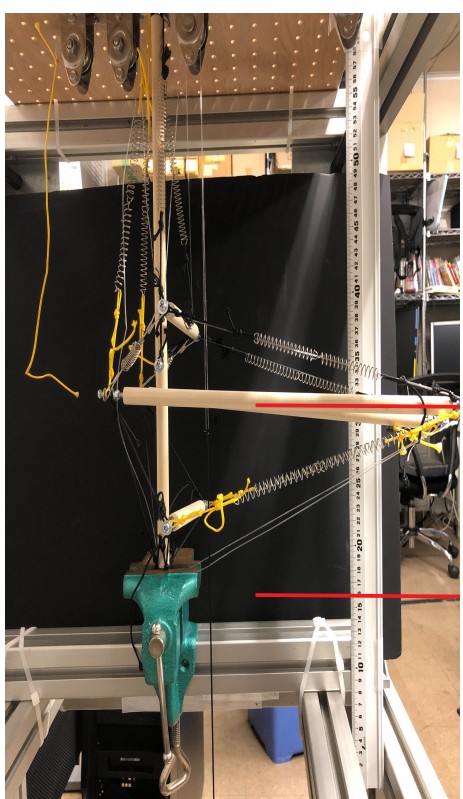

**Figure 14.** The lower arm is being pulled up with a weight of 5000 g. The upper red line is the height of the tip, and the lower red line is the initial height of the tip shown in Figure 10. The tip of the arm is pulled by a weight through the path of the pulley string shown in Figure 11.

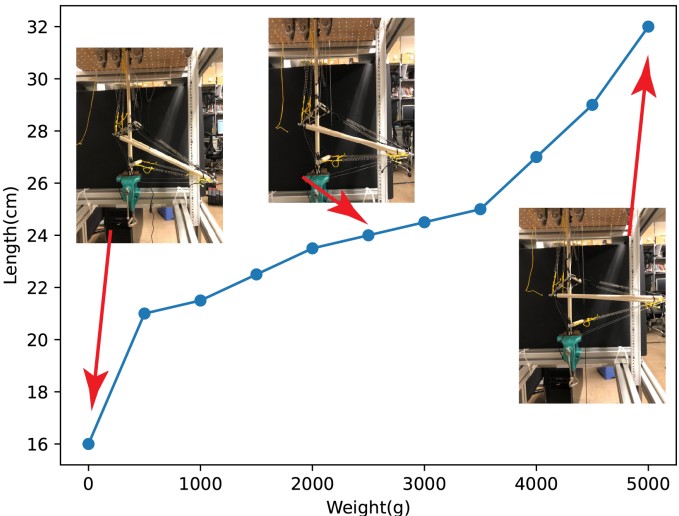

**Figure 15.** Graph of the height of the tip of the lower arm when it is pulled upward by weights. The height of the tip with no weight is 16 cm from the base of the device. The embedded images show the states corresponding to the weights.

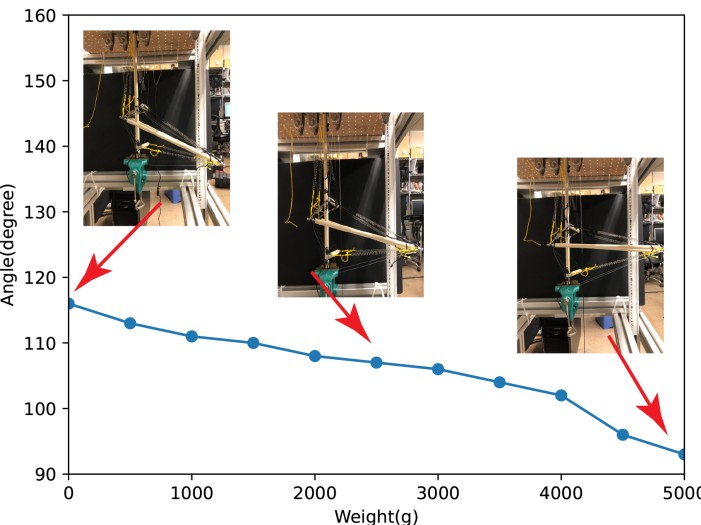

**Figure 16.** Graph of the angle between the upper arm and the lower arm when it is pulled upward by weights. Angles are measured based on images taken from a fixed camera. The embedded images show the states corresponding to the weights.

### 3.2.2. Downward Range of Motion

Figure 17 shows the tip pulled down with a weight of 2500 g. For weights over 3000 g, measurements were stopped because the compression materials came into contact with each other, and the shape of the arm began to distort. Figure 18 shows the graph of the height of the tip, and Figure 19 shows the graph of the angle between the upper arm and the lower arm when it was pulled down by weights.

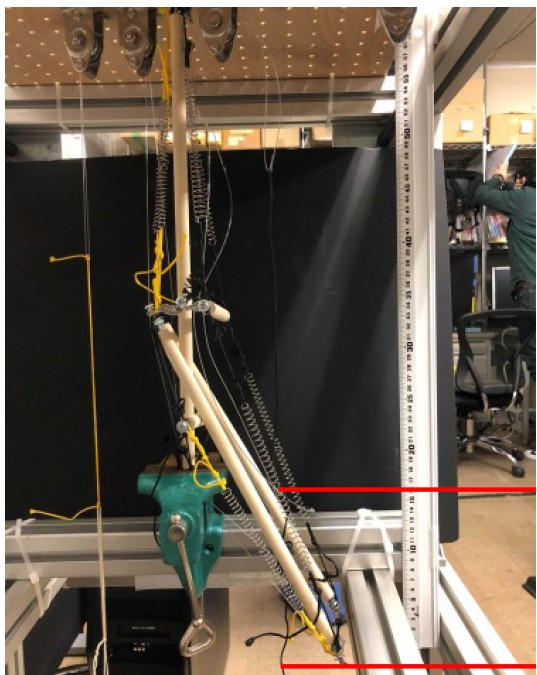

**Figure 17.** The lower arm is being pulled down with a weight of 2500 g. The upper red line is the initial height of the tip shown in Figure 10. The lower red line is the height of the tip. The tip appears lower than its actual height due to the angle of the camera. The tip of the arm is pulled by a weight through the path of the pulley string shown in Figure 12.

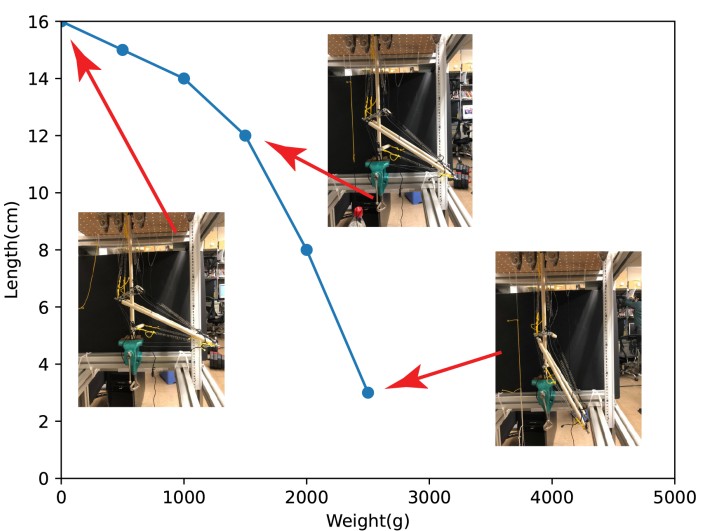

**Figure 18.** Graph of the height of the tip of the lower arm when it is pulled downward by weights. The embedded images show the states corresponding to the weights.The height of the tip with no weight is 16cm from the base of the device. For weights over 3000 g, measurements were stopped because the compression materials came into contact with each other, and the shape of the arm began to distort.

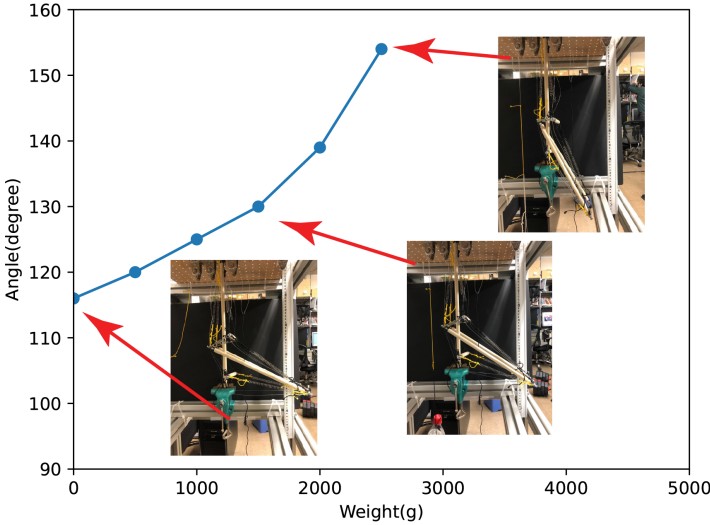

**Figure 19.** Graph of the angle between the upper arm and the lower arm when it is pulled downward by weights. Angles are measured based on images taken from a fixed camera. The embedded images show the states corresponding to the weights. For weights over 3000 g, measurements were stopped because the compression materials came into contact with each other, and the shape of the arm began to distort.

### 3.3. Lifting Ability

Figure 20 shows the graph of the height of the tip with a 500 g weight attached when it was pulled upward by other weights. The height without the 500 g weight on the tip of the lower arm was 17.7 cm. It was able to return above its original height when a 2500 g weight was attached.

### 3.4. Control by Electric Winches

Figures 21 and 22 show how the electric winches control the up and down movements. The electric winch allowed the lower arm to move up and down without problems.

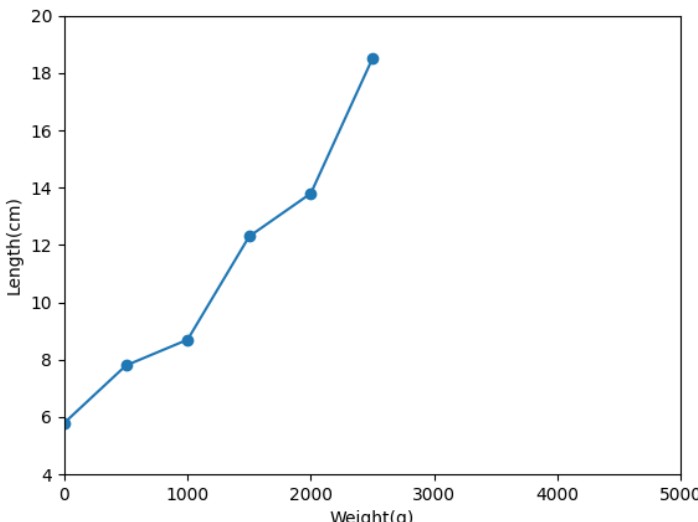

**Figure 20.** Graph of the height of the tip of the lower arm with a 500 g weight attached to the tip of the lower arm when it is pulled upward by the other weights. The height without the 500 g weight on the tip of the lower arm was 17.7 cm.

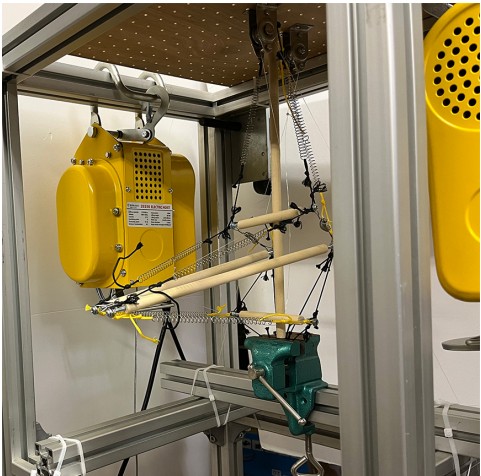

**Figure 21.** The tip of the lower arm is controlled upward by the electric winch.

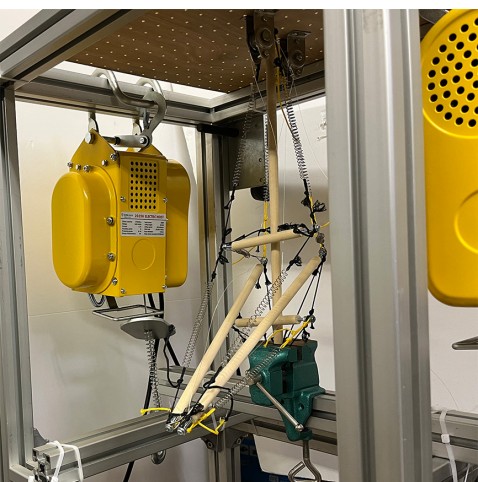

**Figure 22.** The tip of the lower arm is controlled downward by the electric winch.

**4. Discussion**

When a spring is inserted into a tensile material, the length of the tensile material becomes variable, making it difficult to determine the initial length of the tensile material. Therefore, in this study, the lengths of the tensile materials were determined by trial and error. In Section 3.1, we were able to record the lengths of the springs and the strings for each tensile material of the developed arm. The joints were modified to allow the use of springs as part of the tensile materials for the original model. We believe that those who wish to create this tensegrity model may easily do so by referring to this paper.

An average force of 4.16 N was applied to each tensile material in the normal condition of the arm. This means that the tip of the lower arm was pulled from each direction by the forces. We think that these pulling forces may lead to vibration damping at the tip.

As shown in Section 3.2, we were able to find the range of motion of the arm and the force required to do so. By placing the spring in the tensile material, the structure became capable of storing forces of 49.0 N (=5000 g $\times$ 9.8 m/s$^2$, upward) and 24.5 N (=2500 g $\times$ 9.8 m/s$^2$, downward). We consider that creating a mechanism that releases the applied force all at once will lead to the development of technology for arms that move quickly.

The upward folding of the arm only moved to a 93-degree bend, as shown in Figure 16. As shown in Figure 19, the downward opening movement could only be done to 154-degree, and the elbow could not be fully extended. In the case of an actual human arm, the elbow can be bent about 45-degree or 180-degree. In this experiment, the string is used to pull the tip. However, the spring of the individual tensile members needs to be made to contract instead.

As shown in Section 3.3, it was necessary to attach a 2500 g weight to lift a 500 g weight, but it was able to lift it. Having this tensegrity arm simply perform the task of lifting objects may be less force efficient than a normal robot arm because the force is stored in each spring. However, we believe that because force is stored in each spring, it is possible for the movement to change abruptly or release all the force at once. To make such a movement, an actuator with a structure that allows the string to be wound up and then suddenly released is necessary. In addition, to link the movements of each actuator, sensors would be needed to acquire information on the length of the string wound by the actuators and the acceleration of each compression member. As for where to place the actuators and the sensors, we are considering either inside the compression member or outside the compression member where it would not interfere with the tension member.Since there is limited space for making the placement, we are considering using a small motor for the actuator and a small acceleration sensor for the sensor.The amount of hoisting is to be estimated from the input to the motor and the value of the acceleration sensor.By having the actuator repeat the action of winding up the string and then suddenly releasing it, it is also possible to investigate the characteristics of the arm with respect to periodic movements based on the sensor values.

There have been analyses by statics and kinematics for Class 1 tensegrity structure with nodes at both ends of bar compression members [5]. However, our arm has nodes in the middle of the compression member and cannot be analyzed in the same way. Therefore, a structure that is stable without placing nodes in the middle of compression members needs to be the future.

We believe that the mechanism of inserting this spring into the tensile member may lead to the development of a structure that transmits force by flexing, such as a baseball pitcher's arm.

**5. Conclusions**

We modeled an upper and lower arm, proposed a tensegrity arm structure using springs as part of the tensile members, obtained the length of the tensile members through trial and error, and reported the details. We confirmed that the arm had a range of motion that could be bent to some extent by using a simple method to check the motion by pulling



the tip. It was also confirmed that the arm could be controlled by electric winches. We discussed ideas for improvement of the arm, such as connecting all tensile members to both ends of compression members, and the method of attaching the actuators. Furthermore, since the structure of the arm is similar to that of the human body, we believe that future development of the arm will be useful for improving the understanding of how flexible human movements work.

**Author Contributions:** Conceptualization, D.S. and Y.O.; methodology, K.K. and Y.O.; software, K.K. and Y.O.; validation, K.K.; formal analysis, K.K. and Y.O.; investigation, K.K.; resources, D.S. and Y.O.; data curation, K.K. and Y.O.; writing—original draft preparation, K.K. and Y.O.; writing—review and editing, K.K and Y.O.; visualization, K.K. and Y.O.; supervision, D.S. and Y.O.; project administration, D.S. and Y.O.; funding acquisition, D.S. All authors have read and agreed to the published version of the manuscript.

**Funding:** This work was partly supported by JSPS KAKENHI, Grant Number JP19K11428.

**Institutional Review Board Statement:** Not applicable.

**Informed Consent Statement:** Not applicable.

**Conflicts of Interest:** The authors declare no conflict of interest.

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
