# Peer review of "Design of a Movable Tensegrity Arm with Springs Modeling an Upper and Lower Arm"

_actuators, doi:10.3390/act12010018_

Round 1

Reviewer 1 Report

Dear authors,

I find the idea to use tensegrity structures for robot arms very promising. The article presents a real-world model of an arm and the outcomes of experiments conducted with it, which can be valuable.

Unfortunately, as currently drafted  the article has a few significant shortcomings:

·       The introduction needs to be improved significantly;

·       The ‘Materials and Methods’ section is unclear and needs reworking;

·       The results are not presented clearly and cannot be compared to the angles of rotation and work area of, for example, a human arm;

·       There is no conclusion.

Here are some more specific recommendations:

1.     The title does not correspond accurately to the contents. The article does not include details about the control of the structure.

2.     The abstract needs clarify the goal of the research. Perhaps it is to develop a robot arm, similar to the human arm? What kind of purpose could it serve - as a prosthesis; for industrial purposes or other? The result needs to be expressed more clearly (after it has been clarified in the main part of the article).

3.     The opening sentence of the introduction mentions AI in robotics, which has nothing to do with the subject of the article.

4.     The definition of tensegrity can be improved. The principle can be explained in more detail together with a clearer example. In my opinion, [5] (Skelton, R. E.; de Oliveira, M. C. Tensegrity Systems) has good examples.

5.     The references need to be supplemented. For instance: 

a.              https://www.researchgate.net/publication/311680149_Overview_of_tensegrity_-_I_basic_structures 

b.              https://www.angelfire.com/ma4/bob_wb/tenseg.pdf

c.              http://maeresearch.ucsd.edu/skelton/publications/pinaud_mechanics_CRC.pdf

d.              https://www.researchgate.net/publication/355824157_Development_of_a_Modular_Tensegrity_Robot_Arm_Capable_of_Continuous_Bending

e.              https://core.ac.uk/download/pdf/268992342.pdf

Preferably, the references would be between 20-30 and would include more recent research. If available, you should also include the DOI code for the articles. You would need to show what has been done previously in the field of robot arms and what are the advantages of developing a tensegrity arm. You should be more specific about what motivates the work and what results you are looking to achieve.

6.              What type of tensegrity structure are you using? Look at [5] where the different types are classified.

7.              Figures 2-6 are insufficient to describe the structure completely. In my opinion, a 3D CAD model with 2D projections would represent the structure and how the elements are connected more adequately. A side view of the structure needs to show the lengths of the upper arm and the forearm. You could use a fixed coordinate system to describe all dimensions. What are the angles between the wooden sticks? Are the dimensions proportional to a human arm? Clearly indicate which elements are solids, strings and springs.

8.              What kind of motions does the arm do? In section 2.2 (Measurement of range of motion for the tip of the lower arm), figure 7 should show where the tip of the lower arm is. I think it would be clearer if you also include a drawing. Calculate the angles of rotation between the links when accounting for the motion.

9.              Figures 8 and 9 ought to show a 2D side view of the system and include dimensions. You can use numbers to indicate the particular elements of the tensegrity structure.

10.           The methodology needs to be improved and described more clearly. Are trial and error and measurement of deviation the only methods used? This section lacks theoretical justification. Perhaps some formulas from statics and kinematics could be applicable.

11.           The tables do not match the template required by the publishers.

12.           The results are not presented clearly. A figure should indicate the points, between which the lengths represented in Fig 12 and 14 are measured. In my opinion, the trajectories of motion and the ranges of change in the angles need to be shown. The results should be presented in a way that allows comparisons with other similar robot structures or the human arm.

13.           Highlight the advantages of your idea. Perhaps these include greater adaptability, due to the elastic materials used; a lighter structure; and other.

Best regard,

Reviewer

Author Response

Thank you very much for providing important comments. We are thankful for the time and energy you expended. Our responses to the referees’ comments are as follow:

> ·       The introduction needs to be improved significantly;

We have modified the introduction.

> ·       The ‘Materials and Methods’ section is unclear and needs reworking;

An approximate 3D CAD model and descriptions were added to Section 2.

> ·       The results are not presented clearly and cannot be compared to the angles of rotation and work area of, for example, a human arm;

We have added discussion of comparison to the angle of rotation of the human arm.

> ·       There is no conclusion.

Conclusions have added.

> 1.     The title does not correspond accurately to the contents. The article does not include details about the control of the structure.
The title has been changed.

> 2.     The abstract needs clarify the goal of the research. Perhaps it is to develop a robot arm, similar to the human arm? What kind of purpose could it serve - as a prosthesis; for industrial purposes or other? The result needs to be expressed more clearly (after it has been clarified in the main part of the article).
We have added to the abstract and text, for example, that this will lead to research on the use of robots for nursing care and other purposes.

> 3.     The opening sentence of the introduction mentions AI in robotics, which has nothing to do with the subject of the article.
We include AI in the introduction because of our claim that intelligence cannot be separated from embodiment. The introduction has been revised as the claim seems to be difficult to convey.

> 4.     The definition of tensegrity can be improved. The principle can be explained in more detail together with a clearer example. In my opinion, [5] (Skelton, R. E.; de Oliveira, M. C. Tensegrity Systems) has good examples.
We have added explanation of tensegrity in reference to [5]. An explanation was also added to Figure 1.

> 5.     The references need to be supplemented. For instance: 
> a.              https://www.researchgate.net/publication/311680149_Overview_of_tensegrity_-_I_basic_structures 
> b.              https://www.angelfire.com/ma4/bob_wb/tenseg.pdf
> c.              http://maeresearch.ucsd.edu/skelton/publications/pinaud_mechanics_CRC.pdf
> d.              https://www.researchgate.net/publication/355824157_Development_of_a_Modular_Tensegrity_Robot_Arm_Capable_of_Continuous_Bending
> e.              https://core.ac.uk/download/pdf/268992342.pdf
These have been added to the bibliography.

> Preferably, the references would be between 20-30 and would include more recent research. If available, you should also include the DOI code for the articles. You would need to show what has been done previously in the field of robot arms and what are the advantages of developing a tensegrity arm. You should be more specific about what motivates the work and what results you are looking to achieve.
We have added several recent researches to the bibliography.

> 6.              What type of tensegrity struture are you using? Look at [5] where the different types are classified.
It is class 1 in [5], but this is not a typical Class 1 tensegrity structure because there are connections from the middle of a compression members to tension members. We have added about it.

> 7.              Figures 2-6 are insufficient to describe the structure completely. In my opinion, a 3D CAD model with 2D projections would represent the structure and how the elements are connected more adequately. A side view of the structure needs to show the lengths of the upper arm and the forearm. You could use a fixed coordinate system to describe all dimensions. What are the angles between the wooden sticks? Are the dimensions proportional to a human arm? Clearly indicate which elements are solids, strings and springs.

An approximate 3D CAD model was created and placed. If the lengths of compression and tensile members are decided, then the whole structure is uniquely determined. However, a completely accurate 3D CAD model would require time-consuming adjustments to the angles of each material, so an approximate model was created here. The material and length of each element is described in sections 2.1 and 3.1. The lengths of the upper and lower arms are set to values close to those of human arms.

> 8.              What kind of motions does the arm do? In section 2.2 (Measurement of range of motion for the tip of the lower arm), figure 7 should show where the tip of the lower arm is. I think it would be clearer if you also include a drawing. Calculate the angles of rotation between the links when accounting for the motion.

The lines for measurement positions are added to the figures. The angle graphs have also been added.

> 9.              Figures 8 and 9 ought to show a 2D side view of the system and include dimensions. You can use numbers to indicate the particular elements of the tensegrity structure.
Figures 8 and 9 show the paths of the pulleys, so we have added other figures showing 2D side views.

> 10.           The methodology needs to be improved and described more clearly. Are trial and error and measurement of deviation the only methods used? This section lacks theoretical justification. Perhaps some formulas from statics and kinematics could be applicable.

In our model, there is a compression member with connection nodes to the tension members in the middle. Therefore, it cannot be analyzed using statics and kinematics as in Skelton and de Oliveira 2009. For future perspectives, we have added an improvement proposal to make all the nodes be at both ends of the compression members.

> 11.           The tables do not match the template required by the publishers.
Tables 1, 2, and 3 have been modified to correspond to the template.

> 12.           The results are not presented clearly. A figure should indicate the points, between which the lengths represented in Fig 12 and 14 are measured. In my opinion, the trajectories of motion and the ranges of change in the angles need to be shown. The results should be presented in a way that allows comparisons with other similar robot structures or the human arm.

Typical examples have been added in Figures 12 and 14. The angle graphs have also been added. We also added a discussion comparing them to a human arm.

> 13.           Highlight the advantages of your idea. Perhaps these include greater adaptability, due to the elastic materials used; a lighter structure; and other.
The following text have been added.
"The flexibility provided by the springs in the arms is expected to allow the arm to be adaptable and to generate force instantaneously."

Reviewer 2 Report

In the presented manuscript, the authors designed a new structure based on an existing model, and performed a complete evaluation on it. In general, although the novelty is somehow limited, the reviewer agrees with the work, and the manuscript is also solid in terms of language and organization. There are two comment:

1. the reviewer suggests the authors to add a graphical sketch to better illustrate the structure. The real image is good but not clear enough. 

2. the literature review of this paper is short and not enough recent developments are intruduced. There should be more and especially more recent papers.

Author Response

Thank you very much for providing important comments. We are thankful for the time and energy you expended. Our responses to the referees’ comments are as follow:

> 1. the reviewer suggests the authors to add a graphical sketch to better illustrate the structure. The real image is good but not clear enough. 

Based on the results, the CAD-produced figures were added.

> 2. the literature review of this paper is short and not enough recent developments are intruduced. There should be more and especially more recent papers.

We added more reviews of recent papers.

Reviewer 3 Report

The authors present a research on the design and control of a tensegrity arm with springs.

The research is interesting but in its present form, it lacks some consistency on the presentation of the results, on the bibliography, and on the experiments.

+The introduction should emphasize the novelty of the research and the authors should present the organization of the paper.

It is a little bit strange to start the introduction on deep learning and AI, as the control part is not taken into account in this research.

+The references are slightly old.
several researches have been done using tensegrity in robotics, such as the works of Caluwaerts, Valero-Cuevas, Lipson.

A recent review is proposed by DOI: 10.1089/soro.2020.0170

Such works should be cited for instance.

In complement to the reference [14] by Iida on the use of springs for adaptation, you can add
Pitti, Kuniyoshi [2006 and 2010], and for a tensegrity system with springs [Melnyk, Pitti Adv Rob 2018].

+A schematic explanation of tensegrity in Figure 1, should be inserted to explain the force directions with arrows, and the interaction between compressive and tensile elements.
The same for Figure 2, a drawing to explain the different parts of the arm should be inserted to help the reader to understand which parts correspond to the elbow, and forearm.

+For Figures 3 to 6, the authors have to explain the color code and the numbers code in a legend or in the caption text, and in the text.

We don't see clearly where the wires are in Fig10&11, and where the load +is applied

For Fig12, please add another image where there is a super-imposition of the different positions of the end-effector with indications of the corresponding weight (mixture of Fig11 &13 with other positions).

+"gw" is a mistake in page 11?

+I think in its present form the experiment part misses the movement of the tensegrity arm in motion, can the authors make a dynamic experiment with different oscillatory movement.

+An important aspect is in the future, the location and spec of motors embedded within the robotic arm structure. Can the authors explain where they will place them and which types?

Author Response

Thank you very much for providing important comments. We are thankful for the time and energy you expended. Our responses to the referees’ comments are as follow:

> +The introduction should emphasize the novelty of the research and the authors should present the organization of the paper.

We have added text emphasizing the novelty and the organization of the paper in Introduction.

> It is a little bit strange to start the introduction on deep learning and AI, as the control part is not taken into account in this research.

We include AI in the introduction because of our claim that intelligence cannot be separated from embodiment. The introduction has been revised as the claim seems to be difficult to convey.

> +The references are slightly old.
> several researches have been done using tensegrity in robotics, such as the works of Caluwaerts, Valero-Cuevas, Lipson.
>
> A recent review is proposed by DOI: 10.1089/soro.2020.0170

Their papers have been added to the bibliography.

> Such works should be cited for instance.

> In complement to the reference [14] by Iida on the use of springs for adaptation, you can add Pitti, Kuniyoshi [2006 and 2010], and for a tensegrity system with springs [Melnyk, Pitti Adv Rob 2018].

We have added the papers to the bibliography and mentioned them in the text.

> +A schematic explanation of tensegrity in Figure 1, should be inserted to explain the force directions with arrows, and the interaction between compressive and tensile elements.

We responded by editing Figure 1 and adding text.

> The same for Figure 2, a drawing to explain the different parts of the arm should be inserted to help the reader to understand which parts correspond to the elbow, and forearm.

Figure 2 has been modified.

> +For Figures 3 to 6, the authors have to explain the color code and the numbers code in a legend or in the caption text, and in the text.

We have added to the caption and the text.

> We don't see clearly where the wires are in Fig10&11, and where the load +is applied

We have added explanations in the captions of Fig 10,11,13.

> For Fig12, please add another image where there is a super-imposition of the different positions of the end-effector with indications of the corresponding weight (mixture of Fig11 &13 with other positions).

The images of the states corresponding to the weights are inserted in Figures 12 and 14.

> +"gw" is a mistake in page 11?

gw meant grams by weight. The unit was changed to Newton.

> +I think in its present form the experiment part misses the movement of the tensegrity arm in motion, can the authors make a dynamic experiment with different oscillatory movement.

We proposed the following experimental method in Section 4.
"By having the actuator repeat the action of winding up the string and then suddenly releasing it, it is also possible to investigate the characteristics of the arm with respect to periodic movements based on the sensor values."

> +An important aspect is in the future, the location and spec of motors embedded within the robotic arm structure. Can the authors explain where they will place them and which types?

The following was added to Section 4.
"As for where to place the actuators and the sensors, we are considering either inside the compression member or outside the compression member where it does not interfere with the tension member. Since the space available for placement is small, we are considering using a small motor for the actuator and a small acceleration sensor for the sensor."

Round 2

Reviewer 1 Report

Dear authors,

Thank you for taking my comments and recommendations into consideration.

I can see that efforts have been made to improve the article.

The figures still do not clarify the construction. I understand that the structure is spatial, which makes it difficult to present clearly. However, two projections can be given, a frontal view and a view from above. Something similar to the attached Fig.1, to more clearly show what is the tensegrity configuration of rigid body and what are the elastic elements and threads?

Table 1. is unclear. Please edit what is:

"Round wooden stick diameter 12, 40cm"? The reader must guess.

Maybe 12[mm] diameter and 40[cm] length?

It is recommended to use the same units just millimeters, or just centimeters.

The work is still not well motivated. What is the benefit? As far as I understand the movement is realized with the help of significant forces even without the hand being loaded. Is this necessary?

All my comments are only for the purpose of improving the work.

Best regards,

Reviewer

Author Response

Thank you very much for providing important comments again. We are thankful for the time and energy you expended. Our responses to the referees’ comments are as follow:

> The figures still do not clarify the construction. I understand that the structure is spatial, which makes it difficult to present clearly. However, two projections can be given, a frontal view and a view from above. Something similar to the attached Fig.1, to more clearly show what is the tensegrity configuration of rigid body and what are the elastic elements and threads?

We have added the figures. Unlike the proposed figures, no detailed dimensions are provided. The information needed when creating this arm will be the length of each compression member and each tension member, so other dimensions are not necessarily needed. In addition, the portion corresponding to the elbow point is not shown because it does not actually exist.

> Table 1. is unclear. Please edit what is:
> "Round wooden stick diameter 12, 40cm"? The reader must guess.
> Maybe 12[mm] diameter and 40[cm] length?
> It is recommended to use the same units just millimeters, or just centimeters.

Items written in millimeters have been changed to centimeters.

> In my opinion, the table should look like this:
An entry was added to Table 1 for the location of the sticks.

> The work is still not well motivated. What is the benefit? As far as I understand the movement is realized with the help of significant forces even without the hand being loaded. Is this necessary?

The following was added to Introducion:
"The flexibility provided by the springs in the arms is expected to allow the arm to be adaptable and to generate force instantaneously. For example, if compression materials are directly connected at joints, the joints are usually weak, and the compression materials may break at the joints when subjected to a strong impact. However, with a tensegrity structure, the tensile material would be adaptive by absorbing the shock, making it more resistant to impact. We also believe it is possible to quickly move the compression member by loosening only one side of the tensile member from both sides connected to the compression member, which are pulled at the same time."

Reviewer 2 Report

The authors have addressed my previous issues. Thank you.

Author Response

Thank you very much for providing important comments.

Figures and text have been added.

Reviewer 3 Report

can be accepted after minor revision.

Author Response

(The authors gave the same response as above.)
